# Living with a diagnosis of Placenta Accreta Spectrum: Mothers' and Fathers' experience of the antenatal journey and the birth

Helena C. Bartels[1], Antje Horsch[2,3], Naomi Cooney[4], Donal J. Brennan[1], Joan G. Lalor[5]*

1 Dept of UCD Obstetrics and Gynaecology, School of Medicine, National Maternity Hospital, University College Dublin, Dublin, Ireland, 2 Institute of Higher Education and Research in Healthcare, University of Lausanne, Lausanne, Switzerland, 3 Woman-Mother-Child Department, Lausanne University Hospital, Lausanne, Switzerland, 4 Placenta Accreta Ireland, Patient support and advocacy group, Dublin, Ireland, 5 School of Nursing & Midwifery, Trinity College Dublin, Dublin, Ireland

* lalorj1@tcd.ie

**Data Availability Statement:** All relevant data are within the paper and its Supporting Information files.

## Abstract

### Objective

Much research into Placenta Accreta Spectrum (PAS) has focussed on the associated maternal morbidity and mortality. However, mothers' and fathers' lived experiences of the aftermath of a diagnosis of PAS up to the birth and beyond has received little attention. Therefore, the aim of this study was to increase our understanding of the psychological consequences of PAS on women and their partners during pregnancy, up to and including the birth.

### Methods

In-depth interviews were conducted with 29 participants; 6 couples were interviewed together (n = 12), 6 couples were interviewed separately (n = 12), and 5 women were interviewed without their partner. Data from the antenatal and intrapartum periods are presented. Couples were eligible for inclusion if they had a diagnosis of PAS within the previous 5 years. An Interpretative Phenomenological Analysis approach was used to gather and analyse data. Virtual interviews were conducted over a 3-month period from February to April 2021.

### Results

Themes emerged relating to two distinct timepoints, the antenatal period and birth. The antenatal period had two main themes: the first antenatal main theme was "Living with PAS", which had two sub-themes: "Lack of knowledge of PAS" and "Experiences of varied approaches to care". The second antenatal main theme was "Coping with uncertainty", which had two sub-themes of "Getting on with it", and "Emotional toll". Relating to birth, two main themes emerged. The first main theme was "A traumatic experience", with three sub-themes of "Saying goodbye", "Experiencing trauma" and the "Witnessing of trauma" (by

**Funding:** This research received funding from the National Maternity Hospital Foundation, a registered charity operating in Ireland. The funders had no role in study design, data collection and analysis, decision to publish, or preparation of the manuscript.

**Competing interests:** The authors have declared that no competing interests exist.

**Abbreviations:** PAS, Placenta accreta spectrum; MDT, Multi-disciplinary team.

fathers). The second main theme which emerged was "Feeling safe in the hands of experts", with two subthemes of "Safety in expert team" and "Relief at surviving".

## Conclusions

This study highlights the significant psychological consequences a diagnosis of PAS has on mothers and fathers, how they try to come to terms with the diagnosis and the experience of a traumatic birth, and how management within a specialist team can alleviate some of these fears.

## Background

Placenta Accreta Spectrum (PAS) is defined as abnormal trophoblast adherence and invasion into the myometrium of the uterine wall, which results in failure of placental detachment after birth [1]. PAS is classified into three grades depending on the degree of placenta adherence or invasion by the International Federation of Obstetrics and Gynaecology (FIGO) classification [1].

PAS is associated with significant maternal morbidity, largely related to major obstetric haemorrhage and surgical complications [2–4]. Women with PAS may experience many challenges, such as a prolonged antenatal hospitalisation, caesarean hysterectomy, admission to an intensive care unit, and the requirement for neonatal intensive care for the infant [5–7]. Therefore, it is not unexpected that women who experience a pregnancy complicated by PAS face significant physical as well as mental challenges during and after pregnancy. Similarly, fathers may be profoundly impacted by the diagnosis, although our database search did not identify any literature addressing the partners' experience of a pregnancy complicated by PAS. However, previous research on birth trauma after post-partum haemorrhage (PPH) including both women and their partners, suggested that partners were also at risk of post-traumatic stress disorder (PTSD) and depression in the aftermath of such an event [8, 9].

To date, a number of studies have started to explore the psychological sequalae a pregnancy complicated by PAS has on women's mental health. Two quantitative survey studies using validated quality of life questionnaires found mental health scores in women with PAS improved minimally from the time of birth up to 2 years post-birth [10], while the other study found women with PAS were significantly more likely to report anxiety/worry (OR 3.77, 95% CI 1.43–9.93), grief and depression (OR 2.45, 95% CI 0.87–6.95), and overall reported decreased quality of life up to 36 months after the birth [11]. Two qualitative studies of interviews with participants with PAS found many common themes, with women experiencing significant challenges with long-term hospitalisation, a fear of dying and feelings of worry [12], fear and uncertainty during the pregnancy, and birth-related trauma [13]. These studies all suggest there are significant emotional and mental sequelae during and long after a pregnancy complicated by PAS, which warrant further exploration. However, none of the aforementioned studies have included the voice of the partner and their unique experience of PAS. Furthermore, it is recommended that women with PAS are managed within a multi-disciplinary team as this is associated with reduced maternal morbidity [4, 7], however, how women and their partners feel knowing they are being cared for within a multi-disciplinary team (MDT) has not been previously explored.

The aim of this study was to increase our understanding of the psychological consequences of PAS on women and their support partners during pregnancy, up to and including the birth.

## Methods

This was an interview-based study where mothers and fathers described their personal experiences of a pregnancy complicated by PAS. This study focussed on the antenatal journey and birth experiences, with the postnatal journey previously described [14]. Detailed methodology was previously published [14]. In brief, couples were invited to describe their lived experience of reacting to the diagnosis of PAS, the management of the pregnancy up to and including the birth, the experience of care provided by a specialist team, and the impact of the experience of PAS on their relationship. An Interpretative Phenomenological Analysis (IPA) was selected as the approach most appropriate because the aim of the study was to understand mothers' and fathers' experiences of a specific phenomenon, namely living with and beyond a diagnosis of PAS [15, 16]. IPA has been used often in research seeking to explore and understand human experiences because of its potential to uncover new concepts and theories of phenomena yet to be described [16].

Feedback on the design was received from Placenta Accreta Ireland (PAI), a patient advocacy group for those who have been impacted by PAS. Involvement of PAI ensured the study was informed by the principles of Patient and Public Involvement (PPI), which was deemed essential for this project. This involved a preliminary meeting with PAI members and researchers to ensure the study design would explore areas of importance to key stakeholders.

Study participants were recruited and was initiated by the PAI lead (NC) following ethical approval. Where women and/or partners expressed an interest to be involved, an information sheet describing the study was sent to participants. Inclusion criteria were as follows: pregnancy complicated by PAS within the past five years, ability to speak English, ability to consent, and age over 18 years. Partners of eligible women were invited to participate. Couples decided amongst themselves whether they preferred to be interviewed together or separately.

Interviews were conducted via Zoom over a three-month period between February and April 2021 by three researchers (HB (obstetrician), JL (midwife), AH (clinical psychologist). The interview guide was developed based on informal interactions with women during PAI support group meetings. The interview guide is included (S1 File); as interviews progressed and new themes developed, this guide was modified to allow further exploration of same. Interview questions were modified depending on whether the couples were interviewed together or separately to allow the emergence of a systemic perspective.

Interviews were recorded using the audio function in Zoom (Zoom Video Communications Inc. 2016) with participant consent and were transcribed using Sonix (Sonix, Inc. San Francisco, 2021). All audio files and transcripts were pseudonymised and allocated a unique study identifier (M = Mother, F = Father) Consent forms were stored on a secure server and could not be linked to the transcripts or audio files to ensure participant confidentiality.

Procedures to ensure methodological rigour and credibility recommended for IPA were followed [15, 16]. To ensure rigour throughout the analytic process, each researcher who conducted the interviews performed an initial analysis of their participants' transcripts. All stages of analyses were discussed at regular meetings of the research team. Themes were developed from data that was prevalent across interviews, important, and substantial. Saturation of themes was defined when refinements did not add any new substantial themes. By this we mean we did not complete our data collection and analyses until no new themes were emerging in the sample to which we had access We undertook peer validation as opposed to participant validation to ensure rigour and the validity of the findings in line with IPA principles. Credibility of the emerging themes was achieved through peer validation with two additional members of the team (DB, NC) who were not involved in data collection.

Themes developed strongly in relation to three distinct episodes of the PAS experience: the antenatal journey, the birth and the postnatal journey. This study will describe the experience from diagnosis up to and including the birth.

Ethical approval was granted by the hospital ethics committee and written informed consent was obtained from participants (ref EC25.2020).

## Results

This study included 29 participants, consisting of 6 couples who were interviewed together (n = 12), 6 couples who were interviewed separately (n = 12), and 5 women interviewed without their partner with a previous diagnosis of PAS. Of those who were interviewed separately, many did so for pragmatic reasons, such as concern for the partner's emotional wellbeing. Others were aware that if both participated together, one partner may defer to the other rather than express their own experiences.

The median (IQR) age was 37.2 (34.1–43.0) and all women were of Irish origin (Table 1). Of note, all women had had at least one previous caesarean birth, with four women having had one previous caesarean birth, and the remainder had two or more. 16 women had a liveborn infant and one experienced a neonatal death. Interviews were conducted between three months to four years following the pregnancy complicated by PAS. The median (IQR) at the time of interview since the PAS pregnancy was 18 months (12.5–21.5). We did not ask participants to remember detail in terms of dates, times, people etc. The interview was very much focussed on their experience of PAS. Unsurprisingly, because of the traumatic experience of PAS, each participant indicated at interview they could recall vividly key aspects of their experience.

Themes which emerged from interviews were focussed on two distinct episodes; the antenatal journey and the birth. There were two main themes relating to the antenatal journey; "Living with PAS" and "Coping with uncertainty". Relating to the birth, there were two main themes; "A Traumatic experience" and "Feeling safe in the hands of the experts" (Fig 1). Superordinate themes with their associated main and subthemes are described below.

### 1. The antenatal journey

**Main theme 1.1: "Living with PAS".** Mothers and fathers described the experience of living with the diagnosis of PAS and managing the pregnancy up to the birth. Participants spoke

**Table 1. Participant demographics.**

|  | Mothers (n = 17) |
| --- | --- |
| Age | 37.2 (34.1–43.0) |
| Gestation at diagnosis (weeks) | 23+0 (20+1–28+5)* |
| Obstetric history (min-max) |  |
| Parity | 2 (1–4) |
| Number of previous pregnancy loss | 0 (0–2)^ |
| Number of living children | 3 (0–5) |
| Length of hospital stay antenatally (days) | 35.4 (0–140.0) |
| Hysterectomy n (%) | 16 (94) |

Data presented as median (IQR) unless otherwise stated

*One mother not diagnosed during pregnancy

^One mother experienced two neonatal deaths and had no living children, 2 other mothers had experienced a previous stillbirth

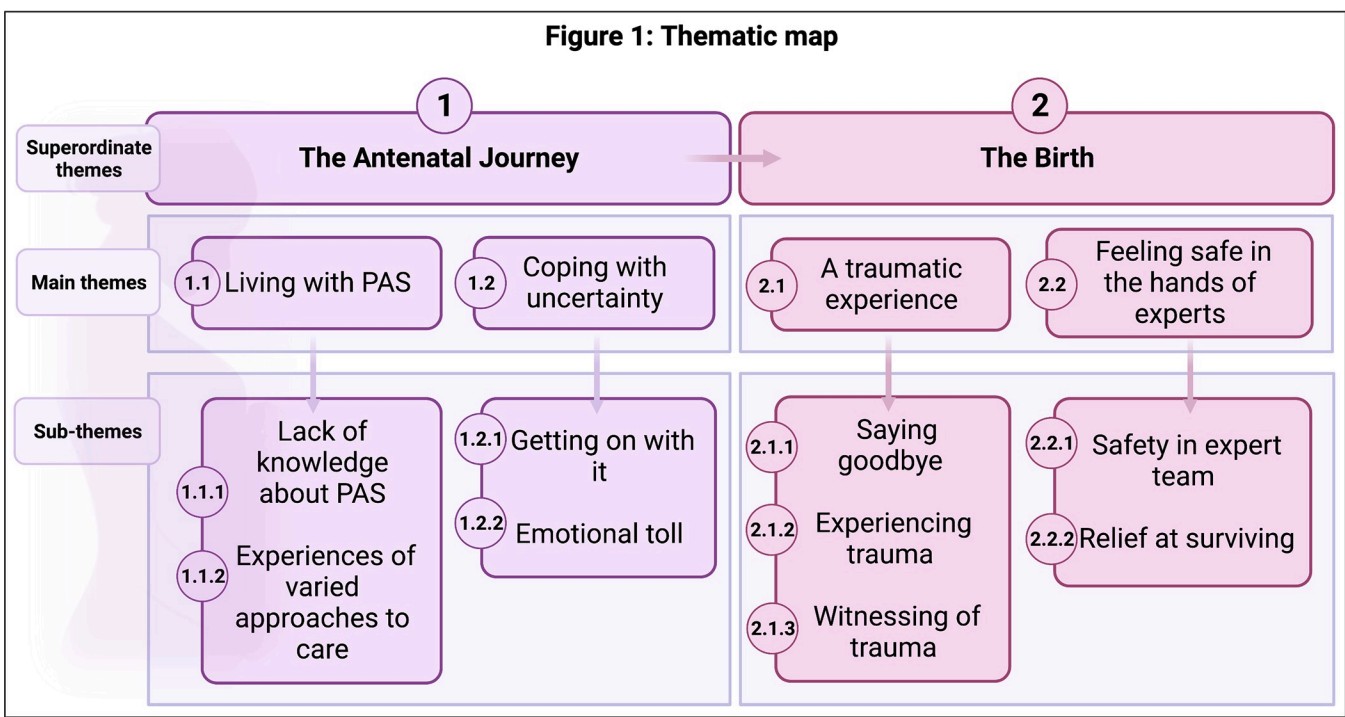

**Fig 1. Thematic map.**

of the fear of a poor outcome, for mothers this related to their self but more so for their baby, and for fathers, they expressed fear for both the life of their partner and for their unborn child. Table 2 provides additional supporting quotes for this theme.

### Subtheme 1.1.1: Lack of knowledge about PAS

Participants reflected on the lack of awareness around PAS. This related to their own lack of knowledge but also for some of the healthcare professionals they engaged with during pregnancy. Both mothers and fathers were shocked at receiving the diagnosis, one father said *"We were shocked. . .we thought we were going in to be given the all clear" (PAS01F)*, only to be told they were suspected of having a condition they were previously completely unaware of. One mother said *"I'd never, ever heard the terminology" (PAS18M)*, with one father saying *"I had never heard of accreta, I began to try educate myself. . .it was hard to get your head around" (PAS02F)*. Some mothers felt healthcare staff were unfamiliar with caring for women diagnosed with PAS, one mother recalled staff were *"just afraid to even get into the diagnosis. . .they were just terrified to come near me" (PAS20M)*.

Mothers and fathers feeling poorly informed about the condition fed into a cycle of fear of the unknown, and an inability to enjoy pregnancy.

**Subtheme 1.1.2: Experiences of varied approaches to care.** Couples reflected on their experiences of the care they received during pregnancy. The experiences of care differed for those cared for within a specialist team compared to those where no perceived structured care system was in place. Where it was evident to the couple that they were being cared for by a specialist team, participants felt their care was being carefully planned, which made them feel safe, with one mother saying *"I couldn't believe how they looked after me. . .everything was so well planned" (PAS01M)* and one father acknowledging that *"Everyone seemed to know what they*

**Table 2. The Antenatal journey.**

| Main themes | Subordinate themes |
|---|---|
| **1.1 Living with PAS** | **1.1.1 Lack of unknowledge about PAS**<br>*Mother*: *"I was just googling and googling. . . I was sent to (another hospital) for an MRI and I needed to be sedated. . . the hysterectomy was never really spoken about to be honest. . . it wasn't in the back of my mind I didn't think you're gonna have your son tomorrow and a hysterectomy. . . they never said it was life threatening. . ." (PAS14).*<br>*Father*: *"They kept asking me the same questions. . .. . .How did they miss this? They said she had a low lying placenta, how did they miss this. . .I'm so angry" (PAS09F)*<br>*Father*: *"And, you know, there was nowhere to look for information. You know, it was just Google searches and, you know, the different types of information we were getting back in. Well it was hard to get you get your head around what was going to happen"(PAS02F)*<br>*Father*: *"I was doing some research at work trying to find out about accreta . . . there didn't seem to be like a lot of information readily available that was quite clear and concise. It was a learning curve" (PAS17F)*<br>*Father*: *"So I'd like to be educated about a accreta a lot better, like, you know, maybe even if it was before someone was diagnosed or maybe have a meeting with people with a group and to try to understand a bit better." (PAS18F)*<br>**1.1.2 Experiences of varied approaches to care**<br>*Mother*: *". . ..I felt (they) were well prepared. . ..everything was so well planned. . .all the doctors and nurses were so good, I couldn't believe how they looked after me. . .I felt like their family. . .it was incredible how we were treated" (PAS01M)*<br>*Father*: *"You're depending on getting the right guy at the right time in, the right team, in the right setting to make it happen. . . I kinda knew it was serious when they started talking about going to (another hospital) for the birth. . ." (PAS17F)*<br>*Father*: *"They just said listen, we're going to sit tight. . .and that's what they did, just waited" (PAS16M)*<br>*Father*: *"I presumed everything was fine. . ..I was relying on the medical doctors to tell me everything was fine. . ..she had signs she didn't feel things were right. . .sonographers saw something of concern but released her without doing anything. . . it was frustrating, I'm not medically trained so what can you do? . . ." (PAS10F).*<br>*Mother*: *"The body saying that things might not quite be right. . . the patient is telling you everything isn't fine. . .." (PAS10M)*<br>*Mother*: *"I found the MRI scan very difficult and it didn't seem to add anything. . .I never heard back about it" (PAS04M).* |
| **1.2 Coping with uncertainty** | **1.2.1 Getting on with it**<br>*Mother*: *"He just focused on this job and the kids. . ..trying to keep everything going at home" (PAS20M)*<br>*Mother*: *"I suppose I kept kind of thinking, I kept on hoping that actually it wasn't accreta because he said he (the doctor) wasn't 100 percent sure that it was. So I kept thinking about this chance it wasn't. It . . .I kept hoping that it wasn't (accreta). . ." (PAS05M)*<br>*Mother*: *"And I was kind of determined that I didn't want to be defined by the diagnosis. I get up and keep going almost along like a distraction, but like kind of a survival kind of mode" (PAS18M)*<br>*Father*: *"I was desensitised to a lot of it. But certainly I don't think I'd be as emotionally attached to things that are happening at the time. I can kind of separate myself, I had just started a business and I was working two jobs. . ." (PAS17F)*<br>*Father*: *". . .I had my hands full (two small children) and it was an hour and a half to get to the hospital to see her.." (PAS14F)*<br>**1.2.2 Emotional toll**<br>*Mother*: *"There was no joy (in the pregnancy) whatsoever. . ..If I could have seen the future, I never would have had another pregnancy, it changed our lives. . .it changed my relationship with my daughter because I wasn't there" (PAS04M).*<br>*Mother*: *"It was just a shock. . .I couldn't take it all in" (PAS21).*<br>*Mother*: *"I mean, I jeopardized myself and my safety, my children's future. . .not knowing whether you are going to make it" (PAS19M).*<br>*Father*: *"It's a roller coaster of emotions . . . why do we deserve this? Why do we have to go through this? This should be the most joyous time of your life." (PAS15F).* |

*were doing so we were very reassured. . . I knew she was in good hands" (PAS04F)*. It was evident mothers and fathers who were cared for within a specialist team felt secure in their care, with one father saying *"any support we needed, we got" (PAS01F)*

However, for other participants the lack of a clear plan was challenging, particularly for women who attended multiple hospitals during the pregnancy *"I was like a ticking time bomb. . .and there was no continuity or plan. . ..I was between hospitals" (PAS14M).* While many participants were fully informed of the associated risks of PAS, some reported they felt unprepared for how serious the condition was, with one mother saying *". . .they never said it was life threatening" (PAS14M).* While mothers expressed a desire of being told all the associated risks, some fathers found this level of information frightening, one father said *"I remember they said she could die and I was thinking oh don't tell me that, I don't want that level of information. . ." (PAS04F).*

Some fathers regretted they were not more available during the pregnancy to attend with their partner for antenatal care and medical consultations. Many were unable to attend hospital appointments for various reasons, such as managing the home and work commitments. For fathers, this meant they were unable to engage with the medical team, as they tried to manage work and home life *"I had my hands full (two small children) and as a I work away, I didn't get to go to the pregnancy checks" (PAS15F).* For mothers, they were left with the burden of responsibility of relaying the medical information to their partners. This proved difficult where mothers were admitted to hospital for long periods of time, as hospital visits often took place with other children present, and hence open discussions around PAS were not possible, one mother said *"We didn't get to talk much, as he came in with the girls (other children)" (PAS11M).* Both mothers and fathers called for joint consultations with the medical team, where possible.

**Main theme 1.2: Coping with uncertainty.** *Subtheme 1.2.1: Getting on with it.* A sense of having to get on with the pregnancy and with life was evident from interviews. Ultimately, this resulted in couples going into "survival mode", with one mother saying *"when you're put in that situation, you just have to keep going" (PAS20M).* Fathers described the desire to help and support their partner as much as possible *"I was trying to stay strong for her. . ..I was very worried. . ..you want to make everything better and you can't" (PAS03F).* Both mothers and fathers focused on short term milestones and little wins along the way to try and get through the weeks *"when they said it doesn't look any worse (at the scan) I took that as a small victory. . ..I focused on the small things" (PAS19M),* and fathers focussed on trying to get to the next gestational milestone *". . . I was wishing the time away from 20 weeks. . .maybe get to 32 and she might be out of the wood" (PAS15F).*

Participants tried to cope by keeping themselves busy with day-to-day tasks. For mothers who were admitted to hospital, they took on a remote role of trying to manage the household from afar, with fathers taking on additional roles at home to manage in the absence of their partner. Fathers described their managerial and organising role *"It was like daddy-day care, I was worried about keeping the money coming in" (PAS17F).* Fathers struggled to balance maintaining day-to-day normality in the household while being there for their partner in hospital. While some couples were well supported by their family and friends, others felt quite isolated, trying to explain the seriousness of their condition to those close to them. One mother said *"I felt nobody else really understood me, like my friends didn't really understand the seriousness. . .so you're trying to tell people" (PAS20M),* while a father commented *"we were very much alone. . ." (PAS01F).*

*Subtheme 1.2.2: Emotional toll.* As the pregnancy progressed, the emotional toll of trying to cope and get on with things was evident. Attempts at coping were made more difficult for mothers as there were so many unknowns and uncertainties *"You'd have a bleed and you would be petrified" (PAS06M).* This fear of the unknown resulted in a heightened sense of anxiety, as there was no guarantee of a good outcome or whether an emergency birth may be necessary, one mother said *"My anxiety was through the roof. . .I was having really bad attacks"*

*(PAS03M)*. Fathers were worried about their partners and what might happen if their worst fears were realised, one father said *". . . I realise now I'd been in denial as my wife might not survive, I thought I was going home to the other kids to say mam is gone" (PAS23F)*. For some mothers, there was a sense of guilt about the pregnancy and diagnosis, as they felt somehow responsible for having pursued another pregnancy *"I just should never have done this. . .I had guilt the whole way through" (PAS04M)*.

## 2. The birth

Participants described their experience of the birth, where trauma emerged as a central theme, with some relief felt knowing they were being cared for by a specialist team. Table 3 provides supporting quotes to explore this theme.

**Main theme 1: A traumatic experience.** *Subtheme 1.1*: *Saying goodbye*. As the day of the birth arrived, the immediate hours before going to the operating theatre were occupied by worry and a focus on saying goodbyes *"It was daunting walking into the room. . .it hit me I had to say goodbye (to partner). . .walking into the room with all those professionals. . .it was very scary" (PAS21M)*. Mothers shared the coping mechanisms on which they relied to deal with the worry of the impending birth *"I said to him you know what to do if the worst happened. . .. once I had that off my chest I knew he'd know what to do"(PAS19M)*. While some fathers emphasised the importance of having time with their partner before they went to the operating theatre, they also acknowledged the need to let the team start their preparations, with one father saying *"I wanted the experts to be able to do their jobs but I wanted to stay with her until the last moment" (PAS23F)*. As mothers went into the operating theatre to start preparing for the birth, some fathers said a support person who could be with them while they waited outside would have been comforting and a welcome distraction. As fathers waited, the thought of losing both their partner and baby was at the forefront of their minds: *"I thought this is life now, I'll be on my own. . ..everything was falling apart" (PAS23F)*.

*Subtheme 1.2*: *Experiencing trauma*. Mothers described the day of the birth as traumatic. An overwhelming sense of vulnerability in the clinical operating room environment was described by some, with one mother saying *"I remember I was absolutely freezing. . .it felt very cold and clinical lying down. . .. I was so conscious of my body" (PAS21M)*. Mothers were accepting of the medical team having to do their job but found the preparation for the birth long and traumatic *"And I know I have to get it done, but that was very traumatic, and it took ages" (PAS22M)*.

Mode of anaesthesia varied between participants. Some mothers had a combination of neuraxial anaesthesia followed by conversion to general anaesthesia once their baby was born, while others had either primary neuraxial or general anaesthesia for the entire duration of the surgery. When reflecting on the day of the birth, mothers' views on the experience of anaesthesia differed between participants. For those mothers who had their birth under primary neuraxial anaesthesia, some reported regret that they were not asleep *"I wish I was asleep, it was just way too traumatic to put somebody through" (PAS04M)*. However, this was not shared by all, with some mothers reporting regret at not being awake for the birth of their baby, one mother said *"I couldn't speak because of the breathing tubes and it had dried out my throat and I was real coarse and dry. . .So I remember struggling to ask questions about how the baby was. . .they showed me a picture (of the baby), but I couldn't focus. My vision was the floor. And so now when I look back, I hate that that all that was taken away" (PAS18M)*. Furthermore, some participants who were awake for the birth were glad as it allowed them to witness the birth of their baby *"it was great. . ...I was able to see him, I was so happy and relieved" (PAS01M)*. Some mothers experienced pain around the time of birth, with one mother saying *"Towards the end of the surgery, I could feel a lot of pain, it was very difficult" (PAS20M)*. Some mothers struggled to

**Table 3. The birth.**

| Main themes | Subordinate themes |
|---|---|
| **2.1. A traumatic experience** | **2.1.1 Saying goodbye**<br>*Mother*: *"We just have to get through this, I wanted to get down there already. You go through the emotions of saying goodbye"* (PAS18M)<br>*Mother*: *"And I knew like by everybody's body language they were all like, oh my God, this is so bad. I didn't know if I was going to wake up again. …everybody was really solemn and grave"* (PAS06M).<br>*Mother*: *"….I'm going to have a baby, what if I don't survive? I was very emotional, I remember giving my husband the biggest hug in case this is the last one ever"* (PAS11M).<br>*Mother*: *"I was completely paralysed with fear, I handed him my wedding band… that was proper goodbye…if he could have been beside me until I was asleep- the last person I saw was a nurse, I was surrounded by strangers. it made it more scary. I really didn't want to go asleep as I thought I wasn't going to wake up.. the dread was huge"* (PAS22M).<br>*Mother*: *"And I wasn't worried about the baby because she was kind of measuring at five pounds. So I knew she wasn't going to be teeny tiny, you know, and I wasn't worried about her, but it was more so I was worrying about my life, like, am I going to wake up? So when I was going in, at the front of my mind was, am I going to wake up from this?"* (PAS05M)<br>*Father*: *"Both lives like what's going to happen…it was so worrying"* (PAS19F).<br>*Father*: *"The nurse called to the room and gave me my (wife's) rings and said do you want to come out and say goodbye to her? I wanted the experts to be able to do their jobs but I wanted to stay with her until the last moment.. I walked out wondering if that was the last time I'd see her alive… then I was left on my own in that room"* (PAS22F).<br>**2.1.2 Experiencing trauma**<br>*Mother*: *"It was scary and I didn't know what I was facing, and so I didn't understand the environment. I was very upset"* (PAS02M).<br>*Mother*: *"My pain was just unreal… that was the worst thing about it"* (PAS06M).<br>*Mother*: *"….it was a lot more traumatic that I ever thought. My first caesarean section with (daughter) was just a normal one, you know, and it was so straightforward. But that was so different. I was too traumatized to just even interact with the baby"* (PAS04M).<br>*Mother*: *"I didn't expect to have my legs put up in stirrups I was totally exposed until someone covered me in a blanket, I felt so vulnerable"* (PAS11M)<br>**2.1.3 Witnessing trauma**<br>*Father*: *"I saw the amount of blood she lost, the colour drained from her face…they rushed her off and they brought me into a room and I was sat there on my own…. I was just collateral, it was frightening on my own… just sitting there… how am I gonna tell her this (the hysterectomy)? They said they will tell her… she lost all the blood in her body twice over she was close to death, it was so unexpected… it was the most scared I've ever been in my life…"* (PAS01F).<br>*Father*: *"Well, when you see someone that you love and in so much pain, it's horrific and there was nothing you could do for her. She was inconsolable, like, because you want to make everything better and you can't, you have to stay strong. I felt helpless"* (PAS03F).<br>*Father*: *"It was how close she came to not surviving, I saw the amount of blood…she didn't see her near death experience"* (PAS10F).<br>*Father*: *"They told me there was massive blood loss and she is stable. I realised when I got to see her I could see how much blood was lost just from the look on her face, I was quite shocked when I saw the colour of her skin and she was still shaking.. you could tell she had gone through something traumatic … they explained it to me.. this was a massive thing she had gone through, and I kinda broke down a bit"* (PAS15F). |

*(Continued)*

**Table 3.** (Continued)

| Main themes | Subordinate themes |
| --- | --- |
| **2.2 Feeling safe in the hands of the experts** | **2.2.1 Safety in expert team**<br>*Mother*: "I was very fortunate with the care and the surgical team, I know that was excellent. . ." (PAS05M)<br>*Mother*: "I'd be at the mercy of somebody that I didn't know (if I bleed when specialist team not there) it's not that I wouldn't have faith in them, but I mean, it left me scared" (PAS1M).<br>*Father*: "You're depending on getting the right guy at the right time in, the right team, in the right setting to make it happen. . ." (PAS17F)<br>*Mother*: "But I suppose I had faith in the fact that they were there (the team). There was a huge number of people in the theatre at that point" (PAS06M)<br>**2.2.2 Relief at surviving**<br>*Mother*: "My husband came in and I kept asking was the baby ok" (PAS11M)<br>*Mother*: "And, you know, the obviously my own consultant, he came he spent some time with us. I couldn't believe I was alive" (PAS06M)<br>*Mother*: "When I first came around, I was like, warm and fuzzy, I was just so, so happy to be waking up. And I remember trying to talk to (partner) and saying, where is she, how is she? But I couldn't speak because of the breathing tubes and it had dried out my throat and I was real coarse and dry, just physically didn't have the energy. So I remember struggling to ask questions about how the baby was, where she was, what she looked like." (PAS18M). |

manage their pain, with one mother commenting that *"even with the medication"* (PAS19M) the pain and discomfort were a real challenge.

*Subtheme 1.3*: *Witnessing trauma*. As well as mothers experiencing a traumatic birth, fathers also reflected on their experience of witnessing this trauma and how this impacted them. Fathers expressed a sense of shock at seeing their wife for the first time after the birth *"I got to see her I could see how much blood she lost from the look on her face. . .I broke down"* (PAS15F). In one sense fathers felt they had seen the worst part, as they had to witness their partner going through the birth, one father said, *"It was how close she came to not surviving, I saw the amount of blood. . .she didn't see her near death experience"* (PAS10F). Fathers shared the mother's sentiment that everything seemed to take so long, with fathers describing what seemed like endless waiting on the day of the birth to be called into the operating theatre or waiting for news *"it just seemed so long, seemed to take forever"* (PAS04F).

**Main theme 2: Safety in the hands of the experts.** *Subtheme 2.1*: *Safety in expert team*. While all participants shared their fears around the birth, they also felt safe in the hands of the healthcare team where it was evident they were in the care of a specialist team. This was shared by both mothers *"I had to trust the team that they knew what they were doing. . ."* (PAS06M) and fathers *"I was amazed at how many people were there . . ..everyone was so professional and knew what they were doing" (PAS01F)*. In particular, participants who were cared for during the pregnancy by a specialist team reported feeling relieved and safe that that same team was there on the day of the birth. Relying on the need for a specialist team to deliver care, however, also meant some couples expressed worry in the antenatal period in the event an emergency out-of-hours birth was needed and their team would not be available.

*Subtheme 2.2*: *Relief at surviving*. As women were transferred to the recovery area or high dependency units, a sense of relief at surviving the birth was evident. Relief was focussed on self-survival but also on the survival of their baby, and the welcome end to this pregnancy. *"I didn't care about blood transfusions or anything. . .it was just such a relief to open my eyes and be alive, it was surreal" (PAS19P)*. There was a sense of gratitude at having survived the birth and surgery. Fathers described waiting outside the operating theatre for news, with one father recalling the relief of being told by the specialist team that his partner was safe *". . .. The*

*consultant was just so good on a personal level, you need that, she came to see me to explain the surgery. . . thank god they saved her. . ." (PAS10F)*

## Discussion

This study explored the psychological consequences of PAS on women and their support partners during pregnancy, up to and including the birth. We present the unique perspective of fathers and what it is like to be the partner of someone with PAS. Interviews revealed a story which, while being centred around fear and uncertainty, demonstrated the strength and resilience of couples as they tried to cope with the diagnosis, with some highlighting the important role of their specialist care team.

Indeed, the comfort couples felt by being cared for by a specialist team and knowing who would be caring for them throughout pregnancy was a key theme emerging from interviews. There is now considerable evidence that women with PAS should be managed within MDTs in specialist care centres, as this has been shown to reduce maternal morbidity and mortality [7, 17, 18]. However, the impact of MDT care has on mothers' experience of care has not been previously described. Our study showed couples who knew they were being cared for within a specialist team reported a strong sense of safety and reassurance. These couples stood out from interviews compared to those women not cared for within a specialist service, in relation to being informed of the planning and continuity of care during the antenatal journey and the birth. This care seemed to alleviate some of the uncertainties and fears associated with PAS. This was evident throughout the pregnancy but also on the day of the birth, where knowing the right team was there brought relief during an otherwise traumatic day.

Although the incidence of PAS continues to increase, most participants had never heard of PAS prior to receiving the diagnosis. This reflects the finding of a previous survey study, where 85% of women diagnosed with PAS had never heard of the condition [10]. Hence, it is not surprising that most couples found themselves dealing with the diagnosis alone, as their family and friends were also unfamiliar with the condition. The most important predisposing risk factor for PAS is a previous caesarean birth [19], a risk factor which all women in this study had, and yet the majority had never heard of the condition. While PAS remains a rare complication of caesarean birth, it seems important that when counselling women about the risks of caesareans, the potential impact on future pregnancies is discussed. Furthermore, as social isolation is a risk factor for deterioration in mental health in pregnancy [20, 21], associated with a higher risk of depression, anxiety, and poor pregnancy outcomes [22, 23], raising awareness around PAS and thus increasing the understanding within their social networks about the condition may alleviate some of the isolation women with PAS feel.

The impact of a traumatic birth on couples was not insignificant. Women who perceive birth as a threat to their life and/or that of their baby are at risk of childbirth-related post-traumatic stress disorder (CB-PTSD) [24]. Women interviewed were unanimous in their description of the birth as "traumatic", with many referring to the day of the birth as "surgery" rather than the day of their child's birth. While CB-PTSD was not formally assessed as part of this study, it is well established that a traumatic birth can have significant psychological sequalae on not only mothers [25–27], but also on fathers [8, 9]. Two studies which included fathers and assessed the psychological impact of a post-partum haemorrhage (PPH) on couples found both women and their partners had higher rates of depression and PTSD compared to those who did not have a PPH [8, 9]. This highlights the importance of including partners in this type of research, as they are also impacted by the outcome of any pregnancy and birth and can go on to develop CB-PTSD. Certainly, the fathers who participated in this study were keen to have their story heard and, for many, reported this was the first time they had discussed the

events of the pregnancy, as they felt they had to be strong and avoided conversations which they feared may be triggering for their partner.

An interesting finding of this study was the contrast in participants' experiences relating to the mode of anaesthesia. For mothers who had neuraxial anaesthesia, some were grateful to have been awake for the birth to witness the birth of their baby, while others perceived being awake as too traumatic because they felt a loss of dignity and very exposed. This highlights the importance of pre-operative counselling and providing education to women on the possible options for mode of anaesthesia. Currently, there is no consensus on the ideal mode of anaesthesia for women with PAS [28]. There is also no data specific to PAS on rates of postnatal depression or PTSD after different modes of anaesthesia. Given the unique clinical factors in each case of PAS, coupled with the contrasting findings of experience of anaesthesia between participants presented in this study, it appears there is currently no strong evidence to recommended one mode of anaesthesia over another. Women should be adequately counselled as to the available options, and, in conjunction with their specialist team come to the best decision whilst taking into consideration the mitigating clinical factors and patient preferences.

This study has a number of strengths and limitations. PAS is a rare condition, and, to our knowledge, this is the largest qualitative study to explore this topic to date, which strengthens the validity of the developed themes. Furthermore, we present for the first time the voice of the partner in the growing literature surrounding the psychological sequalae of a pregnancy complicated by PAS. Methodology using IPA was considered appropriate, as it is particularly useful when undertaking research with small, homogenous samples. This study is strengthened by the strong PPI input at each stage of the process–study design, recruitment, and review of themes, as well as the multidisciplinary of the research team consisting of an obstetrician, midwife, and clinical psychologist. This enhances the validity of the presented themes and ensures the stories are consistent with the experience of a wider group of patient advocates and survivors.

An important limitation of this study is that all women were recruited through a patient support group, and hence may not reflect the experience of women and couples who do not join such a group. However, our findings are similar to previous research conducted with women with PAS [12, 13]. Furthermore, it is possible where women were interviewed with their partner present that this influenced the nuances of the topics discussed. However, in this sample, differences in the data were not apparent based on whether the couples were interviewed together or separately. When these participants were receiving care, some received specialist PAS MDT care, while others were cared for in a standard care setting. For the former, women in this site were fully informed of as to the risks of PAS including loss of life (30). As maternal outcomes are significantly improved when women are care for by a PAS MDT (4), a recently published clinical care guideline in Ireland has recommended a move to a national PAS MDT (31).

## Conclusion

In conclusion, it is evident that a diagnosis of PAS has a profound impact on both women and their partners. Stories shared of the lived experience of PAS centred on fear and going into survival mode. However, the support of a specialist care team ameliorated some of these challenges. This study adds to the growing literature on the psychological impact of a pregnancy complicated by PAS, and highlights the need for additional professional support to be put in place in centres who care for women with PAS.

1. We would like to thank the participants who took part in this study

2. The authors have no conflicts of interest to declare.

3. All authors who contributed to the manuscript are named authors.

4. JL is Chair of Cost Action CA18211 and AH is a management committee member

## Supporting information

**S1 File. The subjective experience of suffering from Placenta Accreta Spectrum (PAS).**
(DOCX)

## Acknowledgments

We would like to acknowledge the women and their partners who participated in this study.

## Author Contributions

**Conceptualization:** Helena C. Bartels, Antje Horsch, Naomi Cooney, Joan G. Lalor.

**Data curation:** Helena C. Bartels, Antje Horsch, Joan G. Lalor.

**Formal analysis:** Helena C. Bartels, Antje Horsch, Joan G. Lalor.

**Funding acquisition:** Helena C. Bartels, Naomi Cooney, Donal J. Brennan, Joan G. Lalor.

**Investigation:** Helena C. Bartels, Joan G. Lalor.

**Methodology:** Helena C. Bartels, Antje Horsch, Donal J. Brennan, Joan G. Lalor.

**Project administration:** Helena C. Bartels, Antje Horsch, Donal J. Brennan, Joan G. Lalor.

**Resources:** Helena C. Bartels, Naomi Cooney, Donal J. Brennan.

**Software:** Helena C. Bartels, Naomi Cooney.

**Supervision:** Antje Horsch, Donal J. Brennan, Joan G. Lalor.

**Validation:** Helena C. Bartels, Antje Horsch, Joan G. Lalor.

**Visualization:** Joan G. Lalor.

**Writing – original draft:** Helena C. Bartels.

**Writing – review & editing:** Helena C. Bartels, Antje Horsch, Naomi Cooney, Donal J. Brennan, Joan G. Lalor.

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
