## [Decision Letter · Decision Letter 0]

27 Feb 2023

PONE-D-22-25947Living with a diagnosis of Placenta Accreta Spectrum: Mothers’ and Fathers’ experience of the antenatal journey and the birthPLOS ONE

Dear Dr. Bartels,

Thank you for submitting your manuscript to PLOS ONE. After careful consideration, we feel that it has merit but does not fully meet PLOS ONE’s publication criteria as it currently stands. Therefore, we invite you to submit a revised version of the manuscript that addresses the points raised during the review process.

We look forward to receiving your revised manuscript.

Kind regards,

David Desseauve, MD, MPH, PhD

Academic Editor

PLOS ONE

Journal Requirements:

This research was funded by the National Maternity Hospital Foundation.

However, funding information should not appear in the Acknowledgments section or other areas of your manuscript. We will only publish funding information present in the Funding Statement section of the online submission form. 

This research received funding from the National Maternity Hospital Foundation, a registered charity operating in Ireland (CHY 20389 registration number). The funders had no role in study design, data collection and analysis, decision to publish, or preparation of the manuscript.

Reviewers' comments:

Reviewer's Responses to Questions

**Comments to the Author**

1. Is the manuscript technically sound, and do the data support the conclusions?

Reviewer #1: Yes

Reviewer #2: Yes

2. Has the statistical analysis been performed appropriately and rigorously? 

Reviewer #1: N/A

Reviewer #2: N/A

3. Have the authors made all data underlying the findings in their manuscript fully available?

Reviewer #1: Yes

Reviewer #2: Yes

4. Is the manuscript presented in an intelligible fashion and written in standard English?

Reviewer #1: Yes

Reviewer #2: Yes

5. Review Comments to the Author

Reviewer #1: this is a very interesting paper and an original way to address psychological impact of PAS while including the partner that usually is under stress and often overlooked. Many couples have dysfunction in their relationship due to distress experienced not only by patients but by other members of the close family (husband and children)

this paper is very well written, is descriptive and is well discussed.

It is very difficult to produce such a kind of paper especially when it is descriptive.

Some minor errors as on line 137 the age should be >= 18

I would ask about if there was a difference between women interviewed lately(around 5 years after) and those who experienced PAS delivery within few months. I think that time can erase or mask some bad memories and render it more acceptable.

Moreover, it is interesting for differentiate between those interviewed with their husband and those without their husband

is there any difference in the reaction between the group of five without husband interviewed and those interviewed separately from their partner?

this paper underlines the importance of management since the beginning by a multidisciplinary team specialized in PAS. it learns us what is lacking in the management of these patients. Sometimes small and insignificant things for us is major and paramount for patient ( experiencing cold, freezing... this is a period the patient thinks she has to go through because she wants to finish with this delivery...but could be avoided by the PAS team with some attention). Publication of this paper is an important testimony for those dealing with PAS

i congratulate the authors for their incredible work and hope seing it published as soon as possible

Reviewer #2: Thank you to the editor for giving me the opportunity to review this manuscript which presents the results of a qualitative study on the experiences of couples with pregnancies marked by placenta accretion.

This is an ancillary analysis of the same material presented in another publication (https://doi.org/10.1186/s12884-022-04726-8). This material consists of 23 interviews with 29 participants. These participants were 17 women with placenta accreta and 12 partners.

The stated aim of this article is to assess the psychological impact of accretion, whereas the aim of their previous article was to describe the experiences of these women and their partners.

In my opinion, a qualitative study should avoid using the terms "impact assessment" but rather talk about identification and/or understanding.

In the abstract, it seems to me that the authors used the term "Placenta Accreta Syndrome" instead of "Placenta Accreta Spectrum". Please confirm. Some errors in the referencing of studies, lines 408 and 473.

The introduction is clear and does not require any comment from me.

The method is rigorous. It might be interesting to clarify the context of care in Ireland, including how women should be informed of the risks. Indeed, in some countries women are systematically informed of all the risks involved, including the darker ones, with their prevalence, whereas in other countries the information is less structured and the numerical risks are not communicated to parents.

Regarding the results, I think it is important to propose a coding tree (in a figure or a table) that is consistent with the aim of this article.

Here is the current structure:

1. Antenatal

1.1. Living with PAS

1.1.1. Lack of knowledge

1.1.2. Experience of care

1.2. Coping with uncertainty

1.2.1. Getting on with it

1.2.2. Emoional toll

2. Birth

2.1. A traumatic experience

2.1.1. Sayying goodbye

2.1.2. Experienceing trauma

2.1.3. Witnessing of trauma

2.2. Feeling safe in the hands of experts

2.2.1. Safety in expert team

2.2.2. Relief at surviving

I believe that the terms chosen to describe the themes could be more explicit to facilitate the link with the aim of the article.

For example, how does surbordinate themes "1.1.1. Lack of knowledge/information" is related to the stated aim line 109 of assessing the psychological impact of the PAS? On the other hand, the issue of lack of information seems to me to have already been published in the section "What needs to change" (https://doi.org/10.1186/s12884-022-04726-8). The authors should consider renaming this theme "fear of the unknwon" for example. Another theme could be "the reassuring side of care planning". I therefore recommend that the authors revise the names "Lack of knowledge" and "Experience of care" to make them consistent with the aim of their analysis.

It would be interesting to know which themes had been anticipated (and were present in the interview grid) and which themes emerged from the data.

The saturation of themes is defined in the line 159 method but is not addressed in the results. Do the authors think they have reached saturation and can they specify this?

In the discussion section, it would be more explicit to use the words used in the objective line 109 to construct the first sentence of the discussion line 385.

The rest of the discussion is clear and makes important points for clinicians as well.

It would be interesting to discuss the context of care in Ireland and data saturation in the strengths and weaknesses section.

This article meets research standards and adds substantially to the existing literature. I therefore recommend its publication.

6. PLOS authors have the option to publish the peer review history of their article (what does this mean?). If published, this will include your full peer review and any attached files.

Reviewer #1: **Yes: **David Atallah

Reviewer #2: **Yes: **Laurent GAUCHER

---

## [Author Response · Author response to Decision Letter 0]

21 Mar 2023

Dear Prof Desseauve,

Thank you to the reviewers for their comments on our manuscript. Please see below a point-by-point response to their review. 

The funding statement for this research is as follows: 

This research received funding from the National Maternity Hospital Foundation, a registered charity operating in Ireland (CHY 20389 registration number). The funders had no role in study design, data collection and analysis, decision to publish, or preparation of the manuscript.

Kind regards,

Dr Helena Bartels

Journal Requirements:

Thank you for this comment. There is no grant number which goes along with this funding, however we have ensured the wording is the same for the cover letter and metadata at submission. (See below, point 3)

This research was funded by the National Maternity Hospital Foundation.

However, funding information should not appear in the Acknowledgments section or other areas of your manuscript. We will only publish funding information present in the Funding Statement section of the online submission form. 

This research received funding from the National Maternity Hospital Foundation, a registered charity operating in Ireland (CHY 20389 registration number). The funders had no role in study design, data collection and analysis, decision to publish, or preparation of the manuscript.

Thank you for this comment. We have removed the fundings statement from the acknowledgements section, and from the end of the manuscript as requested. We confirm the funding statement as outlined above is correct. We have included this statement in the cover letter. 

Thank you for this comment. I have emailed the address as instructed. We would like to proceed with payment by bank transfer or payment link. 

The ethics statement is in the methods section, and has now been removed from the end of the manuscript. 

Thank you we have now included a reference to supporting information S1 on line 168 in the text, and included a reference to this at the end of the manuscript. 

Review Comments to the Author

Reviewer #1: this is a very interesting paper and an original way to address psychological impact of PAS while including the partner that usually is under stress and often overlooked. Many couples have dysfunction in their relationship due to distress experienced not only by patients but by other members of the close family (husband and children)

this paper is very well written, is descriptive and is well discussed.

It is very difficult to produce such a kind of paper especially when it is descriptive.

Thank you for your helpful and positive comments on our manuscript. 

Some minor errors as on line 137 the age should be >= 18

Thank you for pointing this out, we have updated. 

I would ask about if there was a difference between women interviewed lately (around 5 years after) and those who experienced PAS delivery within few months. I think that time can erase or mask some bad memories and render it more acceptable.

Thank you for this comment. The median (IQR) at the time of interview since the PAS pregnancy was 18 months (12.5–21.5). We did not include anyone who had a PAS pregnancy more than 5 years ago (inclusion criteria within 5 years). Most participants were between 1.5-2.5 years after the birth. We did not observe any differences in the themes between those who were more recent compared to further removed from the birth. Traumatic experiences are by their nature particularly well ingrained in the memory. We did not ask participants to remember detail in terms of dates, times, people etc. It was very much focussed on their experience of PAS. 

We have added to the paper on line 187: “The median (IQR) at the time of interview since the PAS pregnancy was 18 months (12.5–21.5). We did not ask participants to remember detail in terms of dates, times, people etc. The interview was very much focussed on their experience of PAS. Unsurprisingly, because of the traumatic experience of PAS, each participant indicated at interview they could recall vividly key aspects of their experience.”. 

Moreover, it is interesting for differentiate between those interviewed with their husband and those without their husband is there any difference in the reaction between the group of five without husband interviewed and those interviewed separately from their partner?

Thank you for this interesting comment and it is something we considered in the study design and analysis. Many of the women who were interviewed alone choose to do so for very pragmatic reasons, such as childcare arrangements. We did not observe a difference in the themes or interviews between those interviewed together or separately. We did find that some women who were interviewed alone said their partner would not be willing to partake, however the experiences of these women did not appear to be different from those who were interviewed with their partner. Of note, all women who were interviewed alone were still in the relationship and there were no single or separated women interviewed. Our impression is there was no difference, but we did not compare these groups. We have now included this as a limitation on line 479 “Furthermore, it is possible where women were interviewed with their partner present that this influenced the nuances of the topics discussed. However, in this sample, differences in the data were not apparent based on whether the couples were interviewed together or separately.”

This paper underlines the importance of management since the beginning by a multidisciplinary team specialized in PAS. it learns us what is lacking in the management of these patients. Sometimes small and insignificant things for us is major and paramount for patient ( experiencing cold, freezing... this is a period the patient thinks she has to go through because she wants to finish with this delivery...but could be avoided by the PAS team with some attention). Publication of this paper is an important testimony for those dealing with PAS

I congratulate the authors for their incredible work and hope seing it published as soon as possible

Thank you for your supportive and positive comments on our manuscript. 

Reviewer #2: Thank you to the editor for giving me the opportunity to review this manuscript which presents the results of a qualitative study on the experiences of couples with pregnancies marked by placenta accretion.

This is an ancillary analysis of the same material presented in another publication (https://doi.org/10.1186/s12884-022-04726-8). This material consists of 23 interviews with 29 participants. These participants were 17 women with placenta accreta and 12 partners.

The stated aim of this article is to assess the psychological impact of accretion, whereas the aim of their previous article was to describe the experiences of these women and their partners.

In my opinion, a qualitative study should avoid using the terms "impact assessment" but rather talk about identification and/or understanding.

Thank you for this comment. We have now removed the word “impact” from the abstract and throughout the manuscript, and replaced with “consequences”. 

In the abstract, it seems to me that the authors used the term "Placenta Accreta Syndrome" instead of "Placenta Accreta Spectrum". Please confirm. 

Thank you for pointing this out. We have updated to the correct terminology of “Placenta Accreta Spectrum”, as used throughout the rest of the abstract and manuscript. 

Some errors in the referencing of studies, lines 408 and 473.

Thank you for highlighting this, we have now updated these references. 

The introduction is clear and does not require any comment from me.

The method is rigorous. It might be interesting to clarify the context of care in Ireland, including how women should be informed of the risks. Indeed, in some countries women are systematically informed of all the risks involved, including the darker ones, with their prevalence, whereas in other countries the information is less structured and the numerical risks are not communicated to parents.

Thank you for this comment. In Ireland, standard practice (1) would be to fully counsel women of serious risks involved in any medical or pregnancy condition with which they are diagnosed, including the risk of death. We have previously written on this area (2) and it would be considered standard for healthcare providers to fully disclose all risks which a ‘reasonable’ person would wish to know. However, we could not say for certain that each woman was told the risks in a similar manor, as the interviewed participants attended multiple different care providers. We have added a line in the limitations to address this also, line 490” For the latter, women in this site were fully informed of as to the risks of PAS including loss of life”

Regarding the results, I think it is important to propose a coding tree (in a figure or a table) that is consistent with the aim of this article.

Here is the current structure:

1. Antenatal

1.1. Living with PAS

1.1.1. Lack of knowledge

1.1.2. Experience of care

1.2. Coping with uncertainty

1.2.1. Getting on with it

1.2.2. Emoional toll

2. Birth

2.1. A traumatic experience

2.1.1. Sayying goodbye

2.1.2. Experienceing trauma

2.1.3. Witnessing of trauma

2.2. Feeling safe in the hands of experts

2.2.1. Safety in expert team

2.2.2. Relief at surviving

I believe that the terms chosen to describe the themes could be more explicit to facilitate the link with the aim of the article.

For example, how does surbordinate themes "1.1.1. Lack of knowledge/information" is related to the stated aim line 109 of assessing the psychological impact of the PAS? On the other hand, the issue of lack of information seems to me to have already been published in the section "What needs to change" (https://doi.org/10.1186/s12884-022-04726-8). The authors should consider renaming this theme "fear of the unknwon" for example. Another theme could be "the reassuring side of care planning". I therefore recommend that the authors revise the names "Lack of knowledge" and "Experience of care" to make them consistent with the aim of their analysis.

Thank you for these insightful and interesting comments. We agree that the themes could be more descriptively named and we are grateful for your suggestions. In terms of “lack of knowledge”, we have added clarity by expanding the title to “lack of knowledge of PAS”. While we appreciate your comment on fear of the unknown, we feel this is addressed under the theme of “coping with uncertainty”, and under the “emotional toll” section. Whilst women who were booked with the MDT team had reassurances of care, this was not the case for women who did not receive this level of care in smaller regional units. Therefore “experiences of care” theme was titled as such to reflect the wide and varied experiences of care. However we appreciate the comment and have altered this to “experiences of varied approaches to care”. 

We have also included a figure to clearly show the thematic map of superordinate, main and subthemes as suggested (named Figure 1). The coding has been updated as suggested starting with theme 1, then superordinate themes 1.1 etc. (Tables and manuscript updated). 

It would be interesting to know which themes had been anticipated (and were present in the interview grid) and which themes emerged from the data.

Thank you for this comment. There was very limited literature available at the time of the study design and recruitment exploring this topic. Therefore, the authors did not anticipate any themes in particular, as the literature in this areas was largely unknown. An interpretive phenomenological analysis approach was taken to explore these unique and largely unknown experiences, to allow us to uncover new and unknown concepts. 

The saturation of themes is defined in the line 159 method but is not addressed in the results. Do the authors think they have reached saturation and can they specify this?

Thank you for this comment. In a small and rare population, of course we can never be certain of reaching data saturation. However, we did not complete our data collection until no new themes were emerging in the sample to which we had access. Therefore, we have added this as further explanation to the methods on line 161: “Saturation of themes was defined when refinements did not add any new substantial themes. By this we mean we did not complete our data collection or analyses until no new themes were emerging in the sample to which we had access”. 

In the discussion section, it would be more explicit to use the words used in the objective line 109 to construct the first sentence of the discussion line 385.

Thank you for this comment. We have now updated the first sentence of the introduction to reflect the language used in the objective line, line 385 now reads: “This study explored the psychological consequences of PAS on women and their support partners during pregnancy, up to and including the birth.”

The rest of the discussion is clear and makes important points for clinicians as well.

It would be interesting to discuss the context of care in Ireland and data saturation in the strengths and weaknesses section.

Thank you for this comment. We have added the following in relation to context of care in Ireland to the limitations: “When these participants were receiving care, some received specialist PAS MDT care, while others were cared for in a standard care setting. For the former, women in this site were fully informed of as to the risks of PAS including loss of life. As maternal outcomes are significantly improved when women are care for by a PAS MDT, a recently published clinical care guideline in Ireland has recommended a move to a national PAS MDT. ”

Data saturation was addressed on line 161 in methods. 

This article meets research standards and adds substantially to the existing literature. I therefore recommend its publication.

6. PLOS authors have the option to publish the peer review history of their article (what does this mean?). If published, this will include your full peer review and any attached files.

Do you want your identity to be public for this peer review? For information about this choice, including consent withdrawal, please see our Privacy Policy.

Reviewer #1: Yes: David Atallah

Reviewer #2: Yes: Laurent GAUCHER

References

1. Guide to Professional Conduct and Ethics for Registered Medical Practitioners

(Amended). 2019. p. https://www.medicalcouncil.ie/news-and-publications/reports/guide-to-professional-conduct-and-ethics-for-registered-medical-practitioners-amended-.pdf.

2. Bartels HC, Brennan DJ. Authors' response to letter to the editor: Quality of life after placenta accreta spectrum: Should women be informed of mortality risk? The Australian & New Zealand journal of obstetrics & gynaecology. 2021;61(4):E25.

---

## [Decision Letter · Decision Letter 1]

9 May 2023

Living with a diagnosis of Placenta Accreta Spectrum: Mothers’ and Fathers’ experience of the antenatal journey and the birth

PONE-D-22-25947R1

Dear Dr. Bartels,

We’re pleased to inform you that your manuscript has been judged scientifically suitable for publication and will be formally accepted for publication once it meets all outstanding technical requirements.

Kind regards,

David Desseauve, MD, MPH, PhD

Academic Editor

PLOS ONE

Reviewers' comments:

Reviewer's Responses to Questions

**Comments to the Author**

1. If the authors have adequately addressed your comments raised in a previous round of review and you feel that this manuscript is now acceptable for publication, you may indicate that here to bypass the “Comments to the Author” section, enter your conflict of interest statement in the “Confidential to Editor” section, and submit your "Accept" recommendation.

Reviewer #2: All comments have been addressed

2. Is the manuscript technically sound, and do the data support the conclusions?

Reviewer #2: Yes

3. Has the statistical analysis been performed appropriately and rigorously? 

Reviewer #2: Yes

4. Have the authors made all data underlying the findings in their manuscript fully available?

Reviewer #2: Yes

5. Is the manuscript presented in an intelligible fashion and written in standard English?

Reviewer #2: Yes

6. Review Comments to the Author

Reviewer #2: Thank you for considering my reviewand for taking the time to create a thematic map to improve the clarity and understanding of your work. Based on my assessment, I consider that the article is of good quality, contains valuable and new information, and can be published as is.

7. PLOS authors have the option to publish the peer review history of their article (what does this mean?). If published, this will include your full peer review and any attached files.

Reviewer #2: **Yes: **Laurent Gaucher

---

## [Editor Report · Acceptance letter]

11 May 2023

PONE-D-22-25947R1 

Living with a diagnosis of Placenta Accreta Spectrum: Mothers’ and Fathers’ experience of the antenatal journey and the birth 

Dear Dr. Bartels:

I'm pleased to inform you that your manuscript has been deemed suitable for publication in PLOS ONE. Congratulations! Your manuscript is now with our production department. 

Kind regards, 

on behalf of

Dr. David Desseauve 

Academic Editor

PLOS ONE